# Effect of Ball Size on the Microstructure and Morphology of Mg Powders Processed by High-Energy Ball Milling

**Jesus Rios** [1,*] **, Alex Restrepo** [1] **, Alejandro Zuleta** [2] **, Francisco Bolívar** [1] **, Juan Castaño** [1] **, Esteban Correa** [3] **and Félix Echeverria** [1]

1   Centro de Investigación, Innovación y Desarrollo de Materiales—CIDEMAT, Facultad de Ingeniería, Universidad de Antioquia UdeA, Calle 70 No 52-21, Medellín 050010, Colombia; humberto.restrepo@udea.edu.co (A.R.); francisco.bolivar@udea.edu.co (F.B.); juan.castano@udea.edu.co (J.C.); felix.echeverria@udea.edu.co (F.E.)
2   Grupo de Investigación de Estudios en Diseño—GED, Facultad de Diseño Industrial, Universidad Pontificia Bolivariana, Circular 1 No 70-01, Medellín 050031, Colombia; alejandro.zuleta@upb.edu.co
3   Grupo de Investigación Materiales con Impacto—MAT&MPAC, Facultad de Ingenierías, Universidad de Medellín, Carrera 87 No 30-65, Medellín 050026, Colombia; escorrea@udem.edu.co
*   Correspondence: jesus.riosa@udea.edu.co; Tel.: +57-(4)-219-6617

**Abstract:** Commercial powders of pure magnesium were processed by high-energy ball milling. The microstructural and morphological evolution of the powders was studied using scanning electron microscopy (SEM), energy dispersive spectrometry (EDX) and X-ray diffraction (XRD). From the results obtained, it was determined that the ball size is the most influential milling parameter. This was because balls of 1 mm diameter were used after a previous stage of milling with larger balls (i.e., 10 and 3 mm). The powder particles presented an unusual morphology with respect to those observed in the Mg-milling literature and recrystallization phenomena. Moreover, the result strongly varied depending on the ball-to-powder weight ratio (BPR) used during the milling process.

**Keywords:** milling parameters; morphological changes; microstructure; recrystallization; magnesium powders

## 1. Introduction

High-energy ball milling (HEBM) is known to be an economical, simple and efficient process [1] for the production of ultrafine powders [2] and can be used to manufacture nanostructured or amorphous materials [1]. There are various factors that must be considered during the milling process. Of these, the initial characteristics of the powders and the milling parameters are the most important, and they all influence the final product obtained [3]. It was reported that an increase in milling variables, such as the ball-to-powder weight ratio (BPR) and milling speed, has a great impact on process energy, accelerating product formation and phase changes [4,5]. As expected, milling balls also play a determining factor in the efficiency of the process, since depending on the type, size, weight and size distribution of these, the milling process can be dramatically affected [1].

Currently, some researchers are especially interested in the milling of Mg because both the refinement of the grain and the reduction in particle size have managed to enhance its applications thanks to its chemical, physical and mechanical properties are improved [6,7]. Now, the recent development of high-energy ball mills that reach rotational speeds of up to 2000 rpm provides the opportunity to acquire materials with novel characteristics, as reported in a previous paper [8]. Therefore, it is attractive to conduct an Mg-milling study involving high milling speeds.

Mg powders exhibit ductile behavior and dynamic recovery phenomenon [9], which causes refinement problems [10] and particle size reduction [11]. Previous studies have found that in order to achieve an effective comminution process of pure Mg powders, a

transition from ductile fracture mechanisms toward brittle fracture mechanisms must be achieved [6]. As a result, cryogenic milling is the most common method [11]. Likewise, other strategies were also implemented, such as the addition of harder particles [12], which change the breaking mechanism and increases the rate of particle size reduction [6]. However, the disadvantage of using milling agents is the contamination generated by the powders. Gülhan Çakmak and Tayfur Öztürk [11] pre-deformed Mg powders by equal channel angular pressing (ECAP), making the powders tend to be more brittle during ball milling. For this reason, the HEBM process studied is a two-stage that involves high speeds and different ball sizes. During the first stage, the aim is to apply a high degree of plastic deformation with large balls so that Mg powder particles present greater fragility. Thus, in the subsequent stage, and with the use of smaller balls, a high-stress intensity is achieved, allowing finer powder particles to be obtained [13].

In this work, a high-energy ball milling process of magnesium powders with increased energetic conditions was explored through modification of BPR and rotational speed. Additionally, the influence of using ball sizes with three different diameters was evaluated, and the effect of these milling variables on the microstructure and morphology of the milled powders was studied. The results showed that powder particles diminished in size as the rotational speed increased and the milling bodies decreased in size. Nevertheless, there is a critical milling body size that causes the powder particles to agglomerate, increase in size and present recrystallization phenomena. Consequently, it appears that under certain milling parameters (especially ball size), ductile powders may exhibit unusual behavior compared with brittle powders.

## 2. Materials and Methods

Commercial powders of pure Mg (purchased from Tangshan Weihao Magnesium Powder Co., Ltd., Qian An, China) were used as raw material with morphology and particle size distribution, as detailed later in the manuscript. The milling process was performed in a Retsch Emax high-energy ball mill (Retsch, Haan, Germany) using zirconia balls in a 50 mL stainless steel vessel with zirconia coating and filled with n-hexane as a process control agent (PCA). A schematic representation of the milling process is shown in Figure 1. In order to avoid oxidation of the powders, the milling container was handled in an MBRAUN Glove Box (Mbraun, Stratham, NH, USA) under an argon atmosphere, with $O_2$ and $H_2O$ contents lower than 0.5 ppm. About 6 g of Mg powder was milled by a process consisting of two or three stages. All samples were initially processed under the conditions indicated for M1. Samples processed in three stages (M6 and M7) were milled in the second stage with the conditions described for M2. Table 1 and Figure 2 detail the milling parameters of each stage and their respective sequences. The selection of the parameters presented above was chosen based on previous experimentation carried out, which established adequate conditions of time, speed and BPR of the respective milling stages studied.

The morphology and chemical composition of both the as-received powders and the milled material was studied using a scanning electron microscope (SEM) JEOL JSM-6490LV (JEOL Ltd., Tokyo, Japan) equipped with microprobe energy dispersive X-ray (EDX) OXFORD INCAPentaFET-x3. The particle size distribution of milled powders was determined from SEM images and ImageJ software (1.52a, National Institutes of Health (NIH), Bethesda, MD, USA), analyzing the Feret diameter of each particle [13]. In order to ensure representativeness for each milling condition, three low magnification images of milled powders were chosen as measurement sources. The average values of particle size measurements were presented in the results. Approximately 500 mg of as-received powder was dispersed with ethanol and measured by laser diffraction (Master Sizer 2000E) (Malvern-PANalytical, Malvern, UK) to determine its particle size. The microstructural study of as-received and milled powders was assessed by means of X-ray diffraction (Empyream Malvern-PANalytical) (Malvern-PANalytical, Malvern, UK), using CuK$\alpha$ radiation and scanning in the range of 2θ = 10°–90° with a step of 0.01°. The full

width at half maximum (FWHM) of XRD peaks was calculated using X'pert High Score Analysis software and ICSD database (3.0e, PANalytical, Almelo, The Netherlands). The crystallite size was determined from the parameters of the most intense Bragg peak and the Scherrer formula [14]. In order to prevent the influence of the instrumental broadening, a correction was created using a pure silicon crystal (standard) and the Gaussian–Gaussian relationship [14–16].

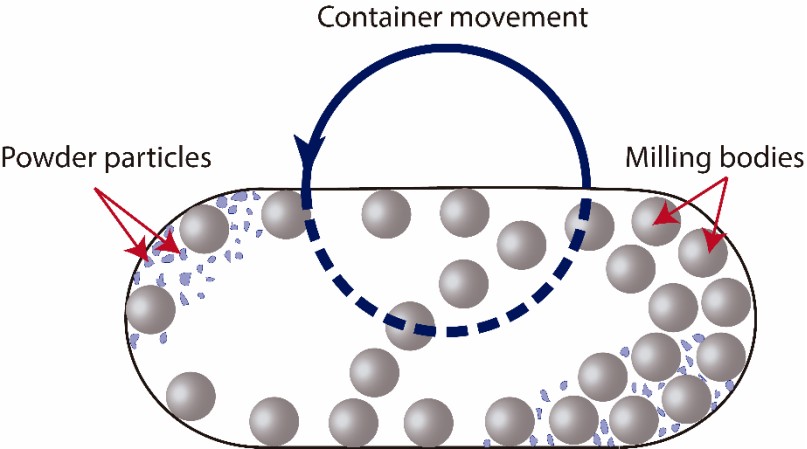

**Figure 1.** Schematic view of the motion of the balls and powder in a vessel during milling process.

**Table 1.** Pure Mg-milling parameters used during the high-energy ball milling stages.

| Sample | Stage | Ball Size (mm) | Rotational Speed (rpm) | BPR | Time (h) |
|--------|-------|----------------|------------------------|------|----------|
| M0 | 0 | - | - | - | - |
| M1 | 1 | 10 | 1000 | 12:1 | 8 |
| M2 | 2 | 3 | 1500 | 11:1 | 4 |
| M3 | 2 | 3 | 1700 | 11:1 | 2 |
| M4 | 2 | 1 | 1700 | 2:1 | 1 |
| M5 | 2 | 1 | 1700 | 1:1 | 2 |
| M6 | 3 | 3 | 1700 | 11:1 | 2 |
| M7 | 3 | 1 | 1700 | 4:1 | 1 |

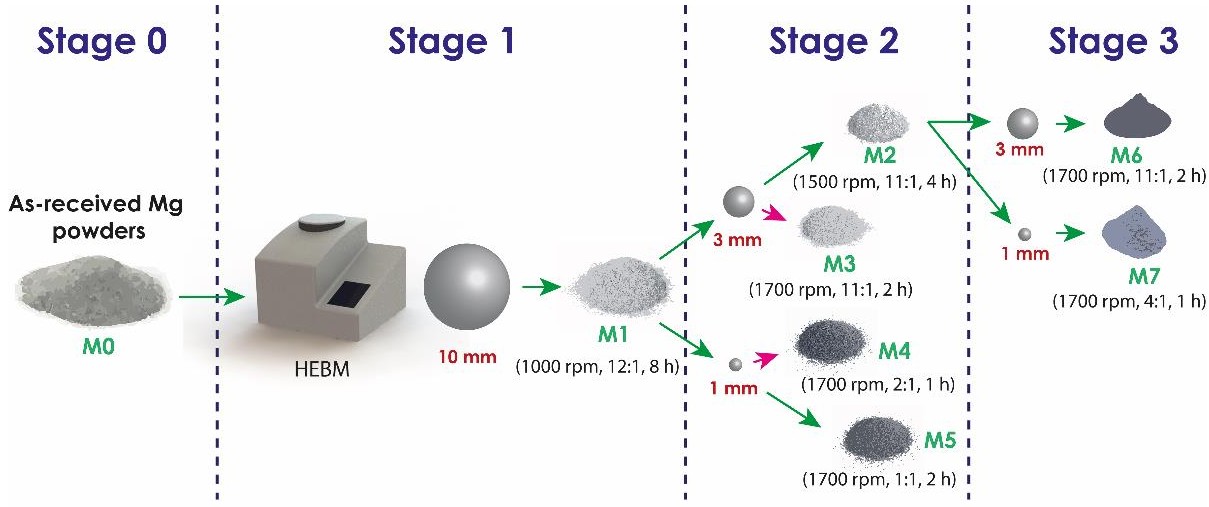

**Figure 2.** Schematic representation of the process stages.

## 3. Results

### 3.1. Morphology of Powders

The chemical composition and morphology of the as-received powder material (M0) and the milled powder labeled as M1 (see Table 1) are shown in Figure 3. It can be observed that the as-received particles exhibit spherical shapes varying from a few microns to tens of microns. These powder particles also have a bimodal size distribution with an average particle size (D50) of 23.73 μm. The EDX analysis of commercial powders (Figure 3b) shows that they have a purity of 95 at.% magnesium and 5 at.% oxygen. Milling of as-received powders for 8 h at 1000 rpm, with 10 mm balls and a BPR of 12:1 (Sample M1), results in an evident morphological transformation (Figure 3c). That is, sample M0 presents a morphological change from spherical particles to plate-like or flake-like particles without uniform sizes (as reported elsewhere [7,17]), ranging from tens of microns to hundreds of microns. Furthermore, there is evidence of particle size growth with a D50 of 53.42 μm. EDX analysis of the milled powders shows the same chemical composition as the starting powders.

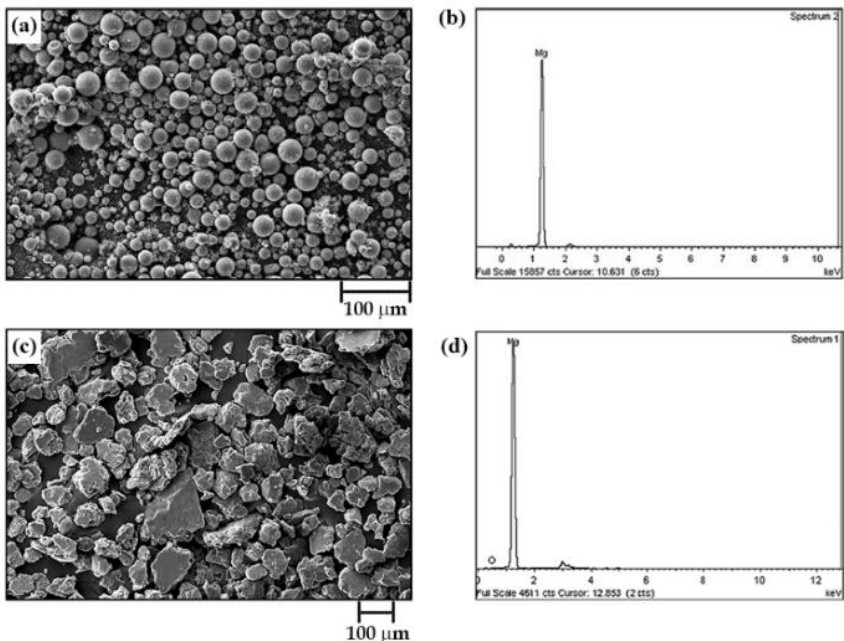

**Figure 3.** Scanning electron micrograph (secondary electrons) and EDX chemical analysis of (**a**,**b**) as-received magnesium powders and (**c**,**d**) sample labeled as M1 (see Table 1).

Figure 4 shows the resulting morphologies of Mg powders following the second and third stages of milling with 3 and 1 mm balls. For milled samples with 3 mm balls (samples M2, M3 and M6), powder particles are finer and slightly flatter when compared to the morphology obtained for sample M1. However, plate-like or flake-like morphology remains after the milling process. The corresponding average particle size (D50) of samples M2 (1500 rpm—11:1—4 h) and M3 (1700 rpm—11:1—2 h) were 31.65 and 29.09 μm. As expected, sample M6 (1700 rpm—11:1—2 h) presented the smallest particle size (D50 = 21.82 μm). Conversely, for samples milled with 1-mm balls (samples M4, M5, and M7), the powders were agglomerated, resulting in a quasi-spherical morphology and a significant increase in particle size. Specifically, the average particle size (D50) of samples M4 (1700 rpm—2:1—1 h) and M5 (1700 rpm—1:1—2 h) was 49.90 and 45.22 μm, respectively. Surprisingly, the D50 of sample M7 (1700 rpm—4:1—1 h) was 292.41 μm. A close inspection of high magnification SEM images (Figure 4g,h) of milled powders reveals that some particles of samples milled with 3 mm balls (Figure 4g) have flat faces and fractured edges, whereas other particles were deformed. Moreover, the observation of powders milled with 1-mm balls (Figure 4h) reveals that several particles were heavily deformed, causing entrapment or encapsulation of other particles.

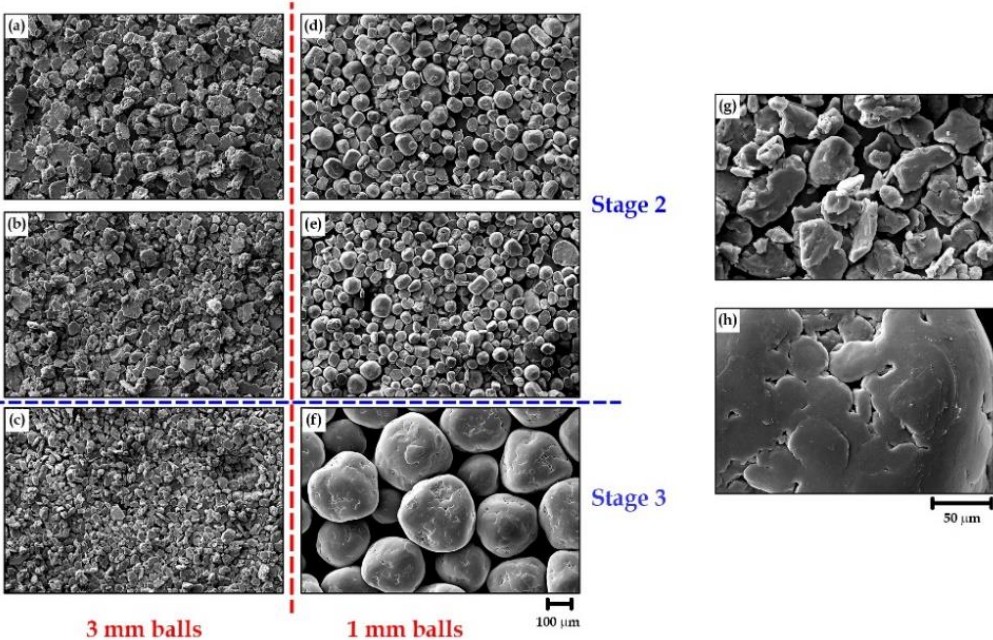

**Figure 4.** Scanning electron micrographs (secondary electrons) showing the resulting morphologies following the second and third stages of milling with 3 and 1 mm balls. (**a**) M2, (**b**) M3, (**c**) M6, (**d**) M4, (**e**) M5 and (**f**) M7. (**g,h**) are enlargements of those in (**c,f**), respectively (see Table 1).

*3.2. Microstructure of Powders*

Figure 5 shows the normalized XRD patterns of the different samples, and Table 2 presents the evolution of crystallite size, lattice strain, dislocation density and lattice parameters derived from the analysis of the XRD results. Even though the X-ray diffraction spectra were collected between 10 and 90° over the 2θ range, only the information corresponding to the range 30–80° is presented in the results, since outside this range, there were no diffraction peaks to report. According to ICSD 98-005-2260 reference standard, all powders presented in Figure 5 show the typical pattern of pure Mg. Nevertheless, all diffraction peaks of milled samples experienced broadening when compared to the peaks of the as-received powders. In addition, the peak corresponding to the plane (002) of milled powder changes in intensity compared to that of the as-received powders (sample M0), with this change depending on the milling stage. Following the first milling stage (sample M1), the intensity of the plane (002) increases. It then begins to decrease during the second milling stage (samples M2 to M5), as smaller diameter balls were used. Finally, in the third stage of milling, when 3 mm balls were used (sample M6), the intensity of the plane (002) is almost equal to that of the previous stage (sample M2); while for the sample milled with 1 mm balls (sample M7), it decreases in intensity almost to that of sample M4. After milling the as-received powders with 10 mm balls (sample M1), the crystallite size of the powders decreased by ~70%, while the lattice strain and dislocation density increased by ~3 and ~12 times, respectively. During the second stage of milling with 3 mm balls (samples M2 and M3), all the parameters mentioned remained almost invariant. However, crystallite size increased by ~1.5 times while lattice strain and dislocation density decreased by ~0.7 and ~2 times, respectively, when the milling of powders was performed using 1 mm balls (samples M4 and M5). Milling of the powders during the third stage, either with 3 mm (sample M6) or 1 mm (sample M7) balls, did not cause significant alterations in crystallite size, lattice strain, or dislocation density. Finally, the lattice parameters "a" and "c" were not affected during any high-energy ball milling process reported here.

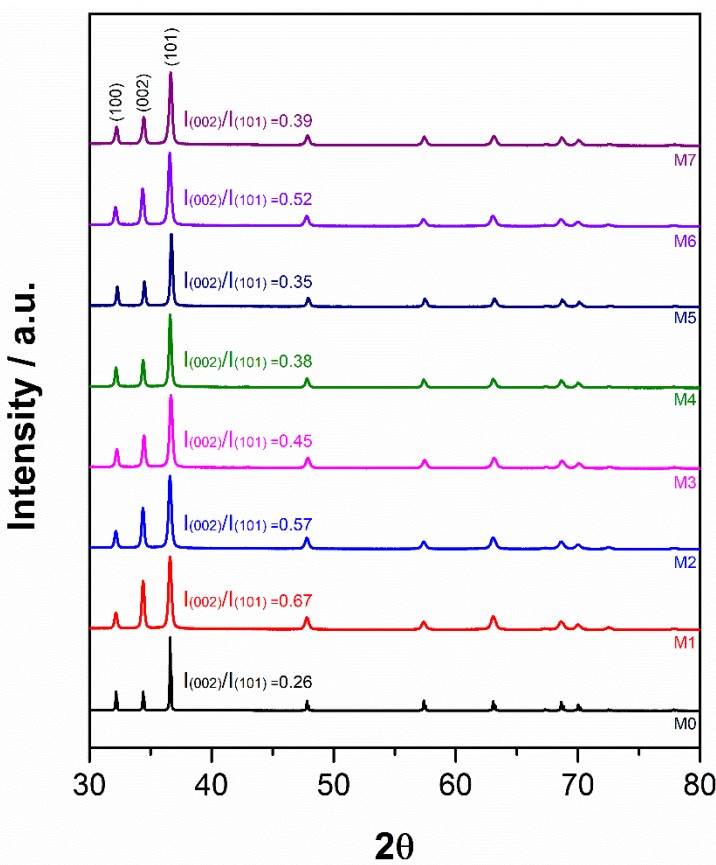

**Figure 5.** XRD patterns of pure Mg powders after high-energy ball milling processes according to the conditions of Table 1.

**Table 2.** Crystal structure data for pure Mg under different milling conditions.

| Sample | Crystallite Size (nm) | Lattice Strain (%) | Dislocation Density ($\times 10^{14}$ m$^{-2}$) | Lattice Parameter (Å) | |
|---|---|---|---|---|---|
| | | | | (a) | (b) |
| M0 | 121.3 | 0.101 | 0.68 | 3.2094 | 5.2110 |
| M1 | 35.0 | 0.326 | 8.16 | 3.2119 | 5.2128 |
| M2 | 35.8 | 0.319 | 7.80 | 3.2117 | 5.2133 |
| M3 | 35.0 | 0.326 | 8.16 | 3.2119 | 5.2128 |
| M4 | 47.0 | 0.245 | 4.53 | 3.2105 | 5.2122 |
| M5 | 51.0 | 0.226 | 3.84 | 3.2061 | 5.2060 |
| M6 | 33.5 | 0.341 | 8.91 | 3.2123 | 5.2145 |
| M7 | 35.1 | 0.324 | 8.12 | 3.2092 | 5.2101 |

## 4. Discussion

The morphological and microstructural evolution of pure magnesium particles varies according to milling parameters. In the first milling stage, and due to the use of 10 mm balls, as-received particles experienced predominantly cold welding, resulting in increased particle size [18,19], while the change in powder morphology and reduction in crystallite size was due to plastic deformation [20–22]. The high impacts produced by these milling bodies generate a high density of dislocations that accumulate in localized regions of the grain, and, as a result, the crystallite is fragmented into small segments of nanometric size [20,23]. The presence of crystalline defects formed during the milling process also increased the lattice strain [18,20]. X-ray diffraction showed that the peak intensity (002) changed during the milling stages. The basal plane (0002) has the same atomic arrangement as the basal plane (0001) [24]. As a result, any changes in Mg (0001) will be reflected in Mg (0002), indicating that deformation occurs through the basal plane sliding, which is common in this material [24–26]. This explains why this increase is mainly more evident

during stage 1 of milling (greater deformation) than in the other stages. As the milling process progresses through the next stages and conditions, grains tend to be randomly oriented, and the plane (002) tends to decrease in intensity [27,28]. Likewise, the sharp decrease in crystallite size exhibited by powders and the increase in the lattice strain caused broadening and, consequently, a decrease in the intensity of diffraction peaks [7,29]. Finally, despite the strong deformation experienced by magnesium powders, this was not enough to modify the HCP structure of pure Mg.

During the second and third stages of milling with 3 mm balls and high speeds, powder particles suffered deagglomeration, hardening and fracturing processes [13,18,19]. In addition, the increase in speed from 1500 to 1700 rpm caused the intensity of stresses generated by the impact, frictional and shear forces (therefore, the amount of energy involved in milling) to be greater, and so the milling time needed for the M3 sample to reach a degree of comminution similar to that of the M2 sample was much shorter (50% less). The use of the third stage with 3mm balls resulted in the M6 sample achieving the smallest particle size due to an increase in accumulated powder damage and milling energy (time and speed). In contrast to the above, the results of milling with 1 mm balls were influenced by parameters such as BPR and time, since the M5 sample (1:1 BPR) needed 1 h more of milling to reach almost the same particle size as that of the M4 sample (2:1 BPR). In other words, the ball-particle interaction was proportional to the number of balls used and the milling time consumed in the comminution process [30]. In general terms, a smaller ball size results in a faster milling process, since in a given volume, both the number of balls and the contact surface increase. This causes the number of collisions between particles and balls to be more intense, resulting in finer and more homogeneous materials [31,32]. The disadvantage of small balls is that their mass is relatively low, and therefore, the im-pact force will be less. This means that smaller milling bodies produce abrasion and attri-tion, which are less efficient breakage mechanisms [33]. For this reason, some authors [30,33] recommend using combinations of ball size, in which larger balls are heavy enough to mill larger and harder particles, while smaller balls are responsible for refining finer particles. However, increasing the milling speed also increases the force of impacts, maximizing the effect of small milling bodies on powder particles [34]. This is especially evident in a high-energy mill such as the Emax used in this study, which can reach rotational speeds up to 2000 rpm [4]. H. Kim et al. [4] studied the influence of the milling body size used in an Emax to mill talc particles (phyllosilicate) at 2000 rpm. Three ball size (2, 1 and 0.1 mm) was used for this purpose, finding that the use of smaller balls did not achieve the same refinement achieved by the other milling bodies. In this study, it is evidenced that the critical size of small milling bodies used in the Emax at high speeds when milling Mg powders could be 3 mm, since, in these conditions, balls of this size may generate enough impact energy to reduce the size of powders. Meanwhile, when using milling bodies with sizes of 1 mm, abrasion and attrition mechanisms were predominant, which eventually caused the unusual behavior of the magnesium powder particles shown in Figure 4 and the changes in crystal structure revealed by XRD results (Table 2).

During the milling process, and despite using liquid PCA to dissipate the heat of milled powders [32], a considerable amount of heat may be released due to the movements of the milling bodies and powder particles, which generate collisions and friction [35]. In this study, the number of milling balls and high rotational speeds allowed the local temperature to influence the milling process since these two factors had a strong influence on both milling efficiency and energy dissipation in the form of heat and plastic deformation [35–37]. When using 1 mm milling bodies, not enough impact force was reached to increase the density of dislocations within the grains of powder particles, and, considering the thermal effect explained above, it was likely that some samples (i.e., samples M4 and M5) suffered stress relaxation processes since Mg is susceptible to such phenomena due to its low melting point [10,28,38]. Additionally, recrystallization and grain growth caused (i) a minimization of internal energy and (ii) a reorganization of the crystals. This occurred mainly in samples milled under the aforementioned conditions

(small milling bodies and high rotational speeds, see Figure 5) [28,39]. Furthermore, the dynamic balance, which sets an equilibrium between the decrease and increase in grain size that occurs due to the different phenomena present in powder milling, allowed the dislocation density, lattice strain and crystallite size to remain almost constant in most samples (samples M2, M3, M6 and M7) [29,38,40].

In our previous paper [8], we found that increasing the milling speed and using 3 mm balls in a relatively short time during titanium milling. Considerable particle size reduction and high powder refinement were achieved. Although Ti and Mg are HCP metals, titanium has a higher melting point and brittleness [8,38]. Which caused the milling efficiency in titanium to be higher than in the present study. Therefore, it is confirmed that Mg milling has certain limitations even though the process conditions are high energy. Additionally, Ka Ram Kim et al. [28] also reported some drawbacks with recrystallization phenomena during a 60 h Mg milling process. On the other hand, in our HEBM process, the crystallite size was obtained around 35 nm. Compared to other studies [10,13], the values are very close, indicating that the minimum crystallite size of pure Mg appears to be around 40–30 nm. The morphologies reported in Mg milling are flat and irregular [11,41]. Nevertheless, under certain milling parameters (especially ball size), other morphologies can be achieved, as evidenced throughout this research. Although cryomilling and milling agents significantly improve the milling efficiency of Mg [12,41]. This process composed of stages could be an alternative without the disadvantages of the previous methods.

## 5. Conclusions

In this study, Mg powders were milled at rotational speeds up to 1700 rpm using a high-energy ball mill. The parameter that most influenced the results was the ball size used in the final stage of the process. Thus, 3 mm balls caused powder particles to undergo deagglomeration and size reduction processes without affecting the microstructural state generated during the first stage. In contrast, 1 mm balls made the powders agglomerate and increase in size until reaching a spherical morphology, accompanied by possible microstructural changes due to the presence of recrystallization phenomena. Consequently, it appears that under certain milling parameters (especially ball size), ductile powders may exhibit unusual behavior compared with brittle powders.

A HEBM process composed of stages is an alternative to obtain pure Mg powders with potential properties for hydrogen storage and structural applications. The possible limitations of the methodology are related to the time consumed in changing the balls and efficiency in reducing the particle size compared to other types of milling. Finally, it would be interesting to study the effect of the morphology of the powders obtained in processes such as additive manufacturing (AM) and powder metallurgy (PM).

**Author Contributions:** Writing—original draft, J.R.; formal analysis, J.R. and A.R.; writing—review and editing, F.E., E.C., A.Z. and J.R.; project administration, F.E.; supervision, E.C. and F.E.; visualization, J.R. and E.C.; funding acquisition, A.Z., F.B., J.C., E.C. and F.E.; resources, F.E. and F.B.; conceptualization, A.R., J.C. and F.B.; methodology, E.C., F.E. and J.R.; investigation; J.R. All authors have read and agreed to the published version of the manuscript.

**Funding:** This research was funded by "Departamento Administrativo de Ciencia, Tecnología e Innovación– COLCIENCIAS (Project 111580862830, contract 183–2019), Universidad de Antioquia, Centro de Investigación para el Desarrollo y la Innovación (CIDI) of the Universidad Pontificia Bolivariana (Rad:482C-05/19–35) and Universidad de Medellín".

**Institutional Review Board Statement:** Not applicable.

**Informed Consent Statement:** Not applicable.

**Data Availability Statement:** Not applicable.

**Conflicts of Interest:** The authors declare no conflict of interest.

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
