# Peer review of "Effect of Ball Size on the Microstructure and Morphology of Mg Powders Processed by High-Energy Ball Milling"

_metals, doi:10.3390/met11101621_

Round 1

Reviewer 1 Report

The logic of the experiment in this paper is clear, the design of the experiment is reasonable. However, the paper lacks of innovation and the introduction of research background is not comprehensive enough

Author Response

The authors appreciate the suggestions made by the reviewer. The following is a point-by-point response to the reviewer's comments.

Point: The logic of the experiment in this paper is clear, the design of the experiment is reasonable. However, the paper lacks of innovation and the introduction of research background is not comprehensive enough.

Response: Amendments to the manuscript were made according to the reviewer suggestion:

1) Possible scientific contributions of the research are mentioned.

2) The following paragraph was added to the introduction to supplement the research background. “Mg powders exhibit ductile behaviour and dynamic recovery phenomenon [9], which causes refinement problems [10] and particle size reduction [11]. Previous studies have found that, to achieve an effective comminution process of pure Mg powders, a transition from ductile fracture mechanisms toward brittle fracture mechanisms must be achieved [6]. As a result, cryogenic milling is the most common method [11]. Likewise, other strategies have also been implemented, such as the addition of harder particles [12]. Which change the breaking mechanism and increases the rate of particle size reduction [6]. However, the disadvantage of using milling agents is the contamination generated to the powders. Gülhan Çakmak and Tayfur Öztürk [11] pre-deformed Mg powders by equal channel angular pressing (ECAP), making the powders tend to be more brittle during ball milling. For this reason, the HEBM process studied is a two-stage that involves high speeds, and different ball size. During the first stage, the aim is to apply a high degree of plastic deformation with large balls so that Mg powder particles present greater fragility. Thus, in the subsequent stage, and with the use of smaller balls, a high stress intensity is achieved, allowing finer powder particles to be obtained [13]”.

Reviewer 2 Report

In the present study, the authors reported the evaluation of the microstructure and morphology of Mg powders prepared by ball milling. The study is addressed to elucidate how the ball size influenced on the final microstructure and morphology of as-prepared Mg powders. In this respect, the authors carefully characterized both commercial and resulting powders. The authors observed that the ratio between ball and powder also played a key role. They prepared several powders by modifying the experimental conditions such as time, weight ratio, etc (Table 1). Despite morphology and microstructure was clearly affected by ball milling treatment, it was observed that the crystalline structure remained unchanged, which is an important aspect to mention from the application point of view. The discussion section has been nicely elaborated. Considering the above mentioned, I would recommend the present manuscript to be published in Metals once the authors modify the manuscript following the minor issues.

My comments are as follows:

- Comment 1. The quality of some figures should be improved (i. e. Labels in Figure 1 are not clear)

- Comment 2. I would recommend the authors to change the nomenclature to make it more clear for the readership.

Author Response

The authors appreciate the suggestions made by the reviewer. The following is a  point-by-point response to the reviewer's comments.

Point 1: The quality of some figures should be improved (i. e. Labels in Figure 1 are not clear)

Response 1: The quality of the images was improved as suggested by the reviewer.

Point 2: I would recommend the authors to change the nomenclature to make it more clear for the readership.

Response 2: According to the reviewer's suggestion, Figure 2 presents a better labelling of the samples to facilitate the reading of the manuscript.

Reviewer 3 Report

Interesting research. Concisely and clearly written.

Suggestions for improvement are as follows:

  1. Avoid group citations. Expand the Introduction section with a detailed overview of previous research. Write for each reference specifically what it is researched.
  2. The manuscript lacks clear critical assessment of the previous research. No clear distinction is made between the previous research and the results obtained in this paper. It is therefore difficult to assess the contribution of this paper; what are the author's findings and what are the distinctions between his claims and the previous work. Do a critical analysis of previous research. Critically and scientifically analyze previous research. Critically highlight the previous researches gap. Point out all the controversies of previous research.
  3. Write the scientific contribution of your research at the end of the Introduction chapter.
  4. Display a figure (photo image) of the milling process.
  5. Further elaborate on the selection of milling parameters (Table 1) Why did you choose these parameter levels.
  6. Potential errors should be discussed.
  7. Estimate the measurement uncertainty of the results.
  8. The Conclusion chapter needs to be significantly supplemented. Additionally write: scientific benefits /scientific contributions of your research; possibilities of practical application in industry; limitations of application of methodology (each research has some shortcomings) and directions of future research.

Author Response

The authors appreciate the suggestions made by the reviewer. The attached document provides a point-by-point response to the reviewer's comments.

Round 2

Reviewer 3 Report

The manuscript has been supplemented and updated.